# Valid and Reproducible Quantitative Assessment of Cardiac Volumes by Echocardiography in Patients with Valvular Heart Diseases—Possible or Wishful Thinking?

**DOI:** 10.3390/diagnostics13071359

**Published:** 2023-04-06

**Authors:** Andreas Hagendorff, Joscha Kandels, Michael Metze, Bhupendar Tayal, Stephan Stöbe

**Affiliations:** 1Department of Cardiology, University Hospital Leipzig, 04103 Leipzig, Germany; joscha.kandels@medizin.uni-leipzig.de (J.K.); michael.metze@medizin.uni-leipzig.de (M.M.); stephan.stoebe@medizin.uni-leipzig.de (S.S.); 2Harrington Heart and Vascular Center, Department of Cardiology, University Hospitals, Cleveland, OH 44106, USA; bhupendar.tayal@gmail.com

**Keywords:** left ventricular volume, echocardiography, cardiac magnetic resonance tomography, mitral regurgitation

## Abstract

The analysis of left ventricular function is predominantly based on left ventricular volume assessment. Especially in valvular heart diseases, the quantitative assessment of total and effective stroke volumes as well as regurgitant volumes is necessary for a quantitative approach to determine regurgitant volumes and regurgitant fraction. In the literature, there is an ongoing discussion about differences between cardiac volumes estimated by echocardiography and cardiac magnetic resonance tomography. This viewpoint focuses on the feasibility to assess comparable cardiac volumes with both modalities. The former underestimation of cardiac volumes determined by 2D and 3D echocardiography is presumably explained by methodological and technical limitations. Thus, this viewpoint aims to stimulate an urgent and critical rethinking of the echocardiographic assessment of patients with valvular heart diseases, especially valvular regurgitations, because the actual integrative approach might be too error prone to be continued in this form. It should be replaced or supplemented by a definitive quantitative approach. Valid quantitative assessment by echocardiography is feasible once echocardiography and data analysis are performed with methodological and technical considerations in mind. Unfortunately, implementation of this approach cannot generally be considered for real-world conditions.

## 1. Introduction

Characterization of cardiac function by echocardiography is predominantly based on volume assessment of cardiac cavities [1,2]. Guidelines and recommendations provide normal values and cut-off values for almost all cardiac diseases [3,4,5]. The calculation of left ventricular (LV) ejection fraction (EF) by biplane planimetry of LV endocardial contours during diastole and systole to estimate LV end-diastolic (LVEDV) and end-systolic volume (LVESV) is performed in almost every patient to characterize systolic LV function [2,6]. Although 2D biplane planimetry to determine LVEDV is recommended in patients with structural heart disease, it was determined in former times by linear measurements using the Teichholz or Quinones method in healthy subjects [7,8]. Further, LVEDV is needed to determine the LV remodeling index as well as the relative wall thickness (RWT) to distinguish between concentric and eccentric types of LV hypertrophy [4,9]. In echocardiography, LV hypertrophy is basically defined by LV wall thickness, RWT, and LV mass (index) [4,6]. In addition to the detection of LV hypertrophy, maximum and minimum indexed left atrial (LA) volumes (LAVI_max_, LAVI_min_) are important parameters to characterize diastolic LV function, especially in patients with heart failure with preserved LVEF (HFpEF) [2].

The volume assessment of right ventricular (RV) volumes by 2D echocardiography is not reliable, due to the triangular shell shape of the RV [10]. Therefore, RV volumes have been characterized by alphanumeric values to analyze patients with right heart diseases since 3D echocardiography became available [11,12]. Right atrial (RA) volumes are determined to characterize RV filling and clinical signs of RV congestion.

Especially in patients with valvular heart diseases (VHD), cardiac volume assessment is important to properly assess the hemodynamic conditions of the individual patient, which is the prerequisite to guide further therapeutic decisions [3,5,13]. A quantitative assessment of cardiac volumes in valvular regurgitations is of particular importance, because relative and absolute regurgitant volumes with corresponding cut-off values are guideline recommendations [1,3,5]. These considerations provide the theoretical basis for the assumption that echocardiography can accurately assess cardiac volumes. While previous studies have not shown significant differences in the determination of LVEF and LV mass between echocardiography and cardiac magnetic resonance (CMR) [14,15], recent meta analyses reported fundamental differences of cardiac volumes between both imaging modalities. In many studies, cardiac volumes determined by 2D/3D echocardiography are basically lower in the ranges of more than 20% compared with CMR as the gold standard [16,17,18,19,20]. In addition, LVEF assessment by echocardiography in comparison with CMR showed both underestimation [21] as well as overestimation [22,23], which can be mainly explained by the limitations of LVEF determination [24].

## 2. Questions and Potential Consequences Arising from These Volume Differences between Echocardiography and CMR

○Do these methodological discrepancies between both imaging modalities really exist?○If LV volumes determined by echocardiography and CMR under comparable circulatory conditions are not similar, how can the problem of underestimation of cardiac volumes by echocardiography or overestimation by CMR be explained?○In addition, if echocardiography is always reputed to measure smaller LV volumes than CMR, how can the data of some previous trials [25], in which comparable or bigger LV volume values have been determined by echocardiography, be explained?○In contrast, if LV volumes determined by echocardiography and CMR under circulatory comparable conditions are similar, are physiological aspects during image acquisition the reason for potential irrelevant differences?○If the assessment of comparable cardiac volumes is possible, does the underestimation of cardiac volumes by echocardiography then constitute a methodological failure?○Assuming that cardiac volumes are underestimated by echocardiography in the past, what are the clinical implications?

The answers to these questions basically offer two main options:(1)both modalities measure the same target values if both methods are used properly;(2)echocardiography generally underestimates cardiac volumes implying that cardiac volumes generally cannot be properly assessed by echocardiography.

While the first option implies that echocardiography offers conclusive diagnostic results by proper cardiac volume analyses, especially for VHD, the second option implies that current echocardiographic recommendations may be based on incorrect cardiac volumes. In consequence, all echocardiographic recommendations—at least for VHD—need to be completely rewritten by defining new normal ranges and cut-off values—particularly for regurgitant volumes and fractions, because both parameters are based on proper LVEDV, LVESV, and RV end-diastolic and end-systolic volume (RVEDV, RVESV) measurements. Furthermore, the second option implies that echocardiography does not generally enable a verifiable quantitative assessment of cardiac volumes.

It is obvious that image acquisition with both modalities—echocardiography and CMR—is often limited in real life. Therefore, this viewpoint focuses on proper and standardized image acquisition using optimized settings which are especially necessary in clinical science. The practical limitations in clinical practice are obvious to introduce a verifiable quantitative approach of cardiac volume assessment. However, this challenge must be accepted to improve echocardiography diagnostics—especially in valvular heart disease.

## 3. Phantom Studies: The True Volumes and the Volumes Determined by Echocardiography Are Comparable

Former phantom studies have shown that quantitative assessment of cardiac anatomy by 2D echocardiography is possible [26]. Recently, phantom studies with symmetrically and asymmetrically shaped volumes as well as experimental studies in animals have shown good agreement in volume determination using 3D echocardiography [27,28,29], gated single-photon emission computed tomography (SPECT) [28], and multi-slice computed tomography (MSCT) [29]. In normal cardiac anatomy, experimental studies have shown that calculated volumes based on echocardiographic measurements correspond to true cardiac volumes, provided that endocardial surface rendering is performed correctly [30]. The blurring of the endocardial surface as a cause of possible measurement differences is due to limited spatial and temporal resolution described in phantom studies [27,31].

All these studies were performed with older ultrasound technology, so that an even better agreement with modern techniques can be assumed. Overall, either no significant or only small volume differences of maximum 1–2% were found compared with the true volume. However, phantom volumes were overestimated by up to 5% using, e.g., MSCT [29]. In conclusion, even with old techniques there are negligible differences between phantom volumes and volumes determined by echocardiography. Therefore, modern echocardiography should be able to determine cardiac volumes validly by a quantitative approach in clinical practice as well as in comparison to, e.g., CMR, provided the acoustic window is sufficiently good.

## 4. Clinical Studies: Different Imaging Modalities Result in Different Cardiac Volumes—Most Notably an Underestimation by Echocardiography Compared with CMR

The frequent reports of underestimated cardiac volume by 2D echocardiography [32,33,34,35,36,37] might be flawed by the use of non-standardized and foreshortened views for planimetry and subsequent calculations [38]. In addition, regional wall motion abnormalities must be taken into account, because, for example, wall motion abnormalities being not visualized in the standardized two- and four-chamber view leads to underestimation of LV volumes and LVEF by calculations based on them [39]. These methodological pitfalls can be avoided by the use of 3D echocardiography—especially real-time 3D echocardiography [30,40]. In addition, poor or insufficient delineation of endocardial contours can be optimized by contrast echocardiography [33,34,36].

Several old as well as recent studies have shown that the determination of cardiac volumes by 3D echocardiography is comparable to CMR, with LV volumes determined by 3D echocardiography having no or only an irrelevant underestimation of less than 10% compared with CMR [14,15,31,40,41,42,43,44,45,46]). In contrast, several studies report a significant underestimation of cardiac volumes by 3D echocardiography compared with CMR [34,35,47]. In this context, an increasing underestimation of LV volumes with the degree of LV dilatation has been reported [48], which could not be confirmed by other studies [45,49]. The wide range of volume differences between measurement by echocardiography and by CMR is shown in many meta-analyses, whereby the more recent studies show smaller or almost no deviations on average especially when using modern real-time 3D echocardiography [16,17,18,19,20].

How to interpret these different findings? Minor volume differences between 3D echocardiography and CMR can be explained by both methodological and physiological causes. Smoothing of regional endocardial irregularities and blurring of the endocardial contour due to low spatial resolution may result in lower cardiac volumes [16,50]. Limited acoustic windows may also cause blurred areas with signal dropouts of the endocardial trabeculae [46]. In addition to these aspects, echocardiographic platforms, analysis software, and underlying automated algorithms generally lead to underestimation of cardiac volumes [51,52], so that vendor specific differences should be considered [51]. An usually lower heart rate of patients during CMR, partly due to the longer preparation and examination time in supine position, means that cardiac volumes determined by CMR might be generally a small amount higher compared with echocardiography [53].

The preliminary summary of these aspects suggests that echocardiography—especially 3D echocardiography due to improvements in modern imaging technology—allows accurate assessment of cardiac volumes. Future reviews about this topic should therefore distinguish between old and new data. Strong discrepancies between cardiac volumes determined by echocardiography and CMR can only be explained by methodological limitations. In general, cardiac volume assessment by 2D echocardiography can only be valid and reliable if measurements were performed in standardized views. Then, even comparability between 2D and 3D echocardiography can be assumed if spatial and temporal resolution of 3D data sets are sufficient, echocardiographic views are standardized, endocardial surface rendering is performed properly (Figure 1), and algorithms of 3D echocardiography reflect the true distances. Clearly, 3D echocardiography is the modality of choice for proper assessment of cardiac volumes under pathological conditions [54,55,56,57,58,59,60].

## 5. Normal Values of Cardiac Volumes in Echocardiographic Recommendations

Despite the knowledge that cardiac volumes can, in principle, be measured just as well by echocardiography as by CMR, the described scenario had an impact on echocardiographic recommendations. Validation studies of cardiac output (CO)/cardiac index (CI) determination by Doppler echocardiography using the velocity time integral (VTI) of the left ventricular outflow tract (LVOT) and the LVOT cross-sectional area compared with CO/CI assessment by thermodilution and the Fick method yield comparable values [61,62,63]. If mean LV stroke volume (LVSV) determined by Doppler echocardiography was approximately 90 mL in the controls [61], LV volumes determined by echocardiographic 2D planimetry, however, were approximately 30% lower in these studies (range of 60 to 70 mL). Interestingly, mean CO/CI values were 5.8 L/min and 3.0 L/min m^2^ for Doppler echocardiography, and 5.0 L/min and 2.6 L/min m^2^ for 2D planimetry, respectively [61].

In most reported normal ranges of LVSV, CO/CI determined by 2D/3D echocardiography in recent recommendations (LVSV approximately 65 mL, CO approximately 4.2 L/min, CI approximately 2.3 L/min m^2^ ) are significantly lower than the comparable ranges determined by Doppler echocardiography) [4,64,65,66,67,68]. Interestingly, normal values determined by CMR are still approximately 15% higher than those determined by Doppler echocardiography (LVSV approximately 110 mL, CO approximately 6.8 L/min, CI approximately 3.8 L/min m^2^) [69], indirectly indicating that the normal ranges of cardiac volumes reported for 2D/3D echocardiography are too low.

The same observations were reported in terms of RV volumes obtained by 3D echocardiography [70,71]. Assuming a normal heart rate of 65–70/min with a given mean RV stroke volume (RVSV) of 57 mL, mean CO and CI values were 4.0 L/min and 2.2 L/min m^2^ [71].

The summary of these reports suggests that current echocardiographic reference may be too low. Therefore, echocardiographic reference values for cardiac volumes need to be critically verified for plausibility, in comparison with other modalities such as CMR. Further, especially for modern 3D echocardiography, the calibrations of the analysis software must be checked with respect to proper delineation of endocardial border. Two-dimensional echocardiography can also serve as a reference in controls, as the mathematical calculations have been validated in normal cardiac geometry [30].

## 6. Implications Derived from the Current Underestimation of Cardiac Volumes by Echocardiography in Patients with Valvular Heart Diseases

Understanding hemodynamics is essentially related to the proper and quantitative assessment of cardiac volumes, CO/CI [1,2,5,72,73,74]. Doppler echocardiography enables the examiner to assess effective CO/CI, if there is no valve regurgitation [75]. In addition, the severity of aortic valve stenosis (AS) is calculated by Doppler echocardiographic measurements using the continuity equation [72] and the prognosis of these patients is estimated by flow conditions defined by LVSV indexed to body surface area (BSA) [76]. Those parameters can only be determined in standardized echocardiographic views with good image quality which allow precise and valid measurements.

Valvular regurgitations complicate the interpretation of hemodynamic measurements. For example, flow conditions in AS patients with combined aortic regurgitation (AR) can be analyzed by the forward LVSV or by the forward RVSV. In this scenario, the effective LVSV and the regurgitant volume of the AR is characterized by the forward LVSV, whereas the forward RVSV represents both the effective RVSV and the effective LVSV, if no pulmonary regurgitation is present (Figure 2). In pure mitral valve regurgitation (MR), the LVSV determined by 2D or 3D planimetry represents the total LVSV, which corresponds to the forward LVSV, which corresponds to the effective LVSV, and the regurgitant volume of the MR (Figure 2). In multiple VHD, the scenario is even more complex when estimating effective LVSV, effective CO, and CI by echocardiography.

In general, in VHD patients, total LVSV is always characterized by 2D or 3D LV planimetry, and the effective LVSV is always characterized by Doppler echocardiography using the VTI of the right ventricular outflow tract (RVOT) or the pulmonary trunk cross-sectional area, respectively, if there is no pulmonary stenosis as well as no or only trace pulmonary regurgitation. The proper LV planimetry in standardized views can objectified by postprocessing in 3D data sets. Thus, foreshortening and consecutive errors of 2D planimetry in non-standardized views can be detected (Figure 3).

Recent recommendations propose an integrative approach to assess the severity of valvular regurgitations [3,5]. In addition to semi-quantitative parameters, e.g., vena contracta and systolic pulmonary vein flow reversal, the error-prone 2D PISA method is recommended as a quantitative method to determine regurgitant volume (RegVol) and regurgitant fraction (RF). For example, severe MR is defined by a RegVol_MR_ ≥ 60 mL and a RF ≥ 50%. Additional information about RegVol_MR_ is “may be lower in low flow conditions” [5]. If a quantitative assessment of these parameters is not possible by echocardiography, switching to CMR is proposed to assess the total and effective LVSV by measuring forward LVSV through the aortic valve as the preferred method for quantification [5].

This approach is understandable because of the reported discordance between echocardiography and CMR in terms of grading MR severity [77,78,79]. The agreement between echocardiography and CMR for grading severe MR using the integrative approach was only 31%, which is ultimately equivalent to or less than chance [80]. These results question the ability to a quantitative assessment of cardiac hemodynamics by echocardiography. This assumption may be true under real-world conditions with limited standardization and limited methodological knowledge about image optimization in echocardiography. However, if echocardiography is performed properly with respect to methodological aspects and standardization, quantitative echocardiography should be possible nowadays.

The question remains if the quantitative assessment of MR severity by echocardiography is based on incorrect small cardiac volumes, where does the general assumption come from that LVEF can be properly assessed if LV volumes cannot be correctly determined? If RF_MR_ determination in echocardiography is predominantly based on 2D PISA method, the effect of underestimating total LVSV is substantial.

A total LVSV of 120 mL is necessary to ensure almost normal CO/CI at normal heart rates of approximately 65/min at a RF_MR_ of 50% (60 mL × 65/min = 3.9 L/min; if BSA is 1.8 m^2^, CI is 2.15 L/min m^2^), if severe MR is defined by a RegVol_MR_ of 60 mL [3,5] according to current recommendations. A total LVSV of 120 mL ultimately requires a LVEDV of up to 400 mL at a LVEF of 30%. If the LV volume were underestimated by 25 or even 50%, the effective LVSV would be 30 mL or even 0 ml, respectively. Even with a falsely low LVEDV of 300 mL and an effective LVSV of 30 mL, the CO and CI would be 1.95 L/min and 1.1 L/min m^2^, corresponding to cardiogenic shock and situations incompatible with daily life [13,81,82,83].

Another thought-provoking aspect of current echocardiographic recommendations is the severity grading of regurgitation based on fixed RegVol especially when the estimation of RegVol is based on results of the 2D PISA method and the total LVSV is ultimately based on questionably low LVEDV and LVESV. If a patient with severely reduced LVEF and normal valve function had a CI of 1.8 L/min m^2^ at the lower limit to the shock index, MR of any severity might be hemodynamically relevant and would result in cardiac decompensation. Thus, even grading by RF_MR_ would be questionable. Consider again the scenario of a true LVEDV of 200 mL with a LVEF of 30% in a patient with optimal medical conservative treatment and a heart rate of 65/min. With a BSA of 1.8 m^2^, CO/CI would be (60 mL × 65/min = 3.9 L/min) 3.9 L/min/2.15 L/min m^2^, respectively. If even mild MR with a RegVol_MR_ of 20 mL develops according to recommendations, the hemodynamic situation corresponds to cardiogenic shock (40 mL × 65/min = CO = 2.6 L/min; CI = 1.45 L/min m^2^). Clearly, myocardial dysfunction is the predominant pathology in this case. However, the cause of acute decompensation is the development of clinically relevant MR, which would have to be classified as formally mild even according to recent recommendations.

## 7. How to Plausibly Analyze the Severity of Mitral Regurgitation—A Case Report (Figure 4)

First, the patient’s baseline data, symptoms, and history must be known and considered.

In the following case, a male 95-year-old asymptomatic patient (height: 172 cm, weight: 69 kg, BSA: 1.8 m^2^, blood pressure: 130/80 mmHg) with sinus rhythm presented to hospital because his daughter, herself a physician, was advised by a cardiologist to undergo interventional mitral valve therapy due to severe MR. The severity in terms of high-grade MR was based on eyeballing of the regurgitant jet area (Figure 2A and Figure 4a(A)) although, e.g., vena contracta is very small and the size of the jet area is not recommended to quantify MR severity [3]. Further, MR should be qualitatively documented and the etiology of MR—in this case, a secondary MR—must be diagnosed. Concomitant valvular defects were excluded, resulting in isolated MR. Ultimately, LVEDV, LVESV, LVEF, and total LVSV must be plausibly determined (Figure 4a(B)—monoplane, Figure 4a(C)—biplane). LVEDV was approximately 150 mL (within normal ranges), LVEF was approximately 60% (within normal ranges), and total LVSV was approximately 90 mL. Normal LVEDV and LVEF already indicate irrelevant MR.

**Figure 4 diagnostics-13-01359-f004:**
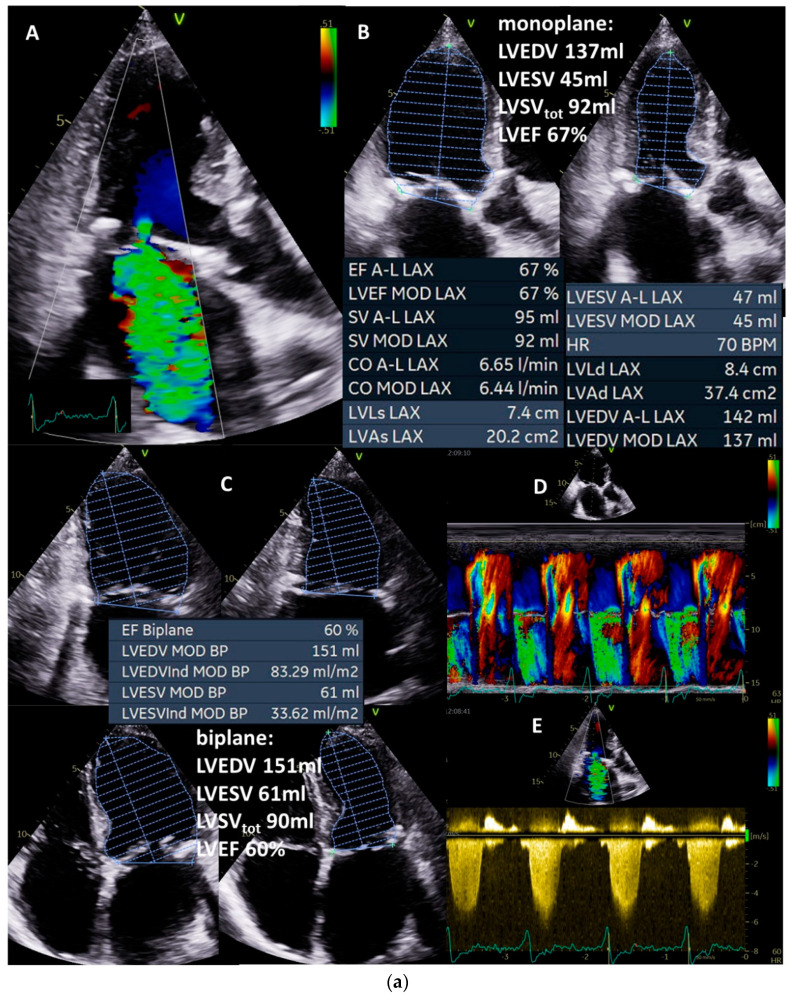
(**a**): Illustration of a systolic regurgitant jet phenomenon in a patient with isolated mitral regurgitation and the quantitative analysis of left ventricular volume parameters. In (**A**), the small vena contracta < 1 mm despite a remarkable jet area is shown. In (**B**), monoplane LV planimetry is shown using the apical long-axis view. In (**C**), the biplane LV planimetry is shown documenting a total LVSV of 90 mL. In (**D**), a color-coded M-Mode through the MR shows a nearly constant PISA radius. In (**E**), the cw regurgitant Doppler velocity is shown. LVEDV = left ventricular end diastolic volume, LVESV = left ventricular end systolic volume, LVSV = left ventricular stroke volume, LVSV_tot_ = total LVSV, and LVEF = left ventricular ejection fraction. (**b**): Illustration to assess quantitatively effective LVSV and RVSV by pw Doppler echocardiography as well as regurgitant volume by 2D PISA method to analyze severity of mitral regurgitation (MR): in (**A**), the parasternal long-axis to document LVOT diameter during systole is shown; in (**B**), the measurements of LVSV_eff_ by pw Doppler spectrum at the level of the LVOT is shown; in (**C**), the parasternal and subcostal short-axis views are shown to document RVOT diameter during systole; in (**D**), the measurements of RVSV_eff_ by pw Doppler spectrum at the level of the RVOT is shown. The estimation of RegVol_MR_ by 2D PISA is demonstrated by delineation of the 2D-PISA radius and the velocity time integral of the retrograde transmitral velocity during systole (**E**,**F**). LVSV = left ventricular stroke volume, LVEF = left ventricular ejection fraction, LVSV_eff_ = effective LVSV, RegVol = transmitral regurgitant volume, LVOT = left ventricular outflow tract, RVOT = right ventricular outflow tract.

At this point in the examination, the findings should be checked for hemodynamic plausibility. Calculating the minimum effective LVSV to enable a required effective CO and CI at a heart rate of 63/min (corresponding to values of approximately CO = 4 L/min; CI = 2.2 L/min m^2^: 4000 mL/min/63/min = 63 mL) facilitates the interpretation. Thus, the maximum possible RegVol_MR_ to be above shock limit is (90 mL − 63 mL = 27 mL), which corresponds to a RF of 30%.

Next, the effective LVSV should be quantified by Doppler echocardiography (Figure 4b(A,B)—pw-Doppler LVOT, Figure 4b(C,D)—pw-Doppler RVOT), yielding approximately 75 mL (Figure 4b(E,F)). Thus, RegVol_MR_ was 15 mL in this case. RegVol_MR_ can additionally be checked by 2D PISA method if there are no methodological limitations to the use of 2D PISA method (Figure 4a(D)—color-coded M-Mode—MR, Figure 4a(E)—cw-Doppler—MR, Figure 4b(E,F)—2D PISA method). Thus, RF_MR_ was less than 20% in this case. In conclusion, secondary alterations, e.g., increased E/e’ ratio and/or increased systolic pulmonary arterial pressure, associated with relevant MR, should be considered to exclude high-grade MR.

In summary, this example documents mild functional MR confirmed by clinical presentation without symptoms, normal ranges of LVEDV and LVEF, and individual mild-grade RF_MR_ < 20% at a calculated effective CO and CI of 4.7 L/min and 2.6 L/ min m^2^, respectively.

The reliability of the echocardiographic volume measurements in this case of a pure mitral regurgitation can be counterchecked by the determination of effective LVSV and RVSV. In addition, the total LVSV must be the sum of effective LVSV and RegVol_MR_. The calculated RegVol_MR_ can be checked in this case by the determination of RegVol_MR_ by the 2D PISA method.

## 8. Future Implications

This viewpoint aims to stimulate an urgent and critical rethinking of the echocardiographic assessment of patients with VHD, especially valvular regurgitations:(1)The actual, integrative approach is too error prone to be continued in this form. It should be replaced or supplemented by a definitive quantitative approach comparable to CMR.(2)Valid quantitative assessment by echocardiography is feasible once echocardiography and data analysis are performed with methodological and technical considerations in mind.(3)Unfortunately, implementation of this approach cannot generally be considered for real-world conditions.(4)Cardiac volumes, particularly total and effective LVSV and RegVol, should be checked for plausibility and valid hemodynamics.(5)Apparently, incorrect former echocardiographic data of cardiac volume measurements should not be considered in future meta-analyses.(6)New studies about normal ranges of cardiac volumes measured by echocardiography and validated by CMR are necessary.(7)The echocardiographic algorithm for analysis of VHD, particularly valvular regurgitations, should be revised to include hemodynamically valid CO and CI.

## Figures and Tables

**Figure 1 diagnostics-13-01359-f001:**
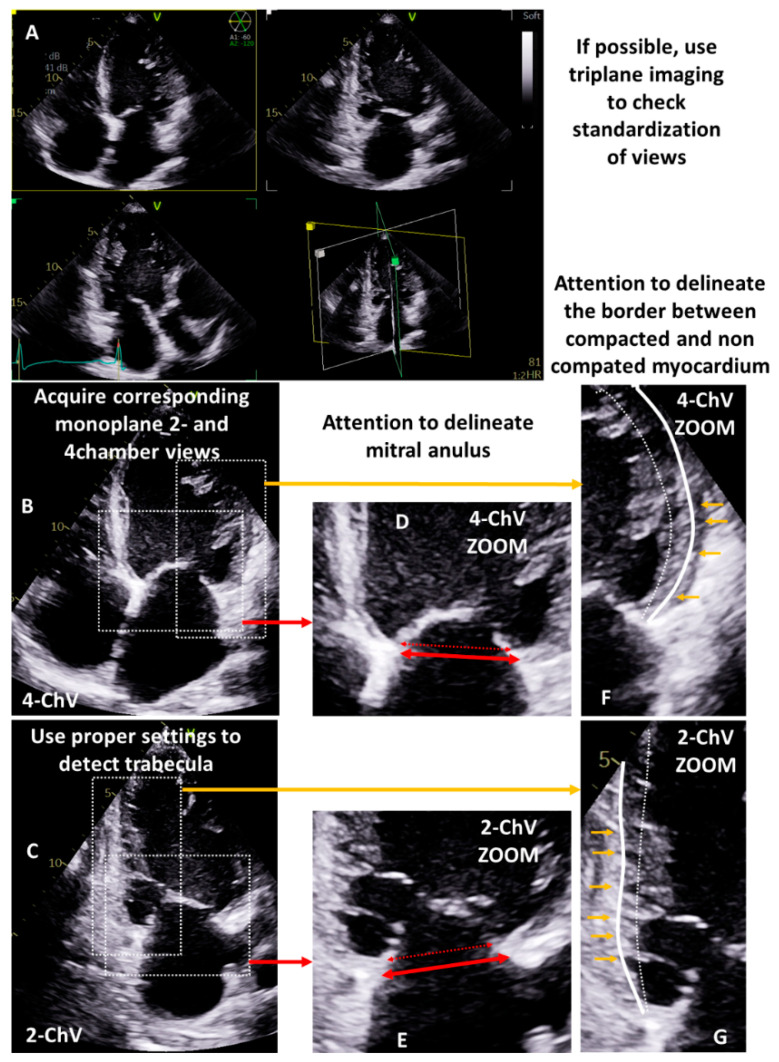
Instructions for proper 2D planimetry or 3D volumetry of the LV cavity. Firstly, check the standardization of apical views by using triplane imaging (**A**). Secondly, acquire comparable monoplane two- and four-chamber views (2-ChV, 4-ChV) with high spatial resolution to ensure proper visualization of cardiac structures (**B**,**C**). Thirdly, properly performed the delineation of mitral anulus—the proximal mitral leaflets normally do not represent the level of mitral anulus (D-Zoom area is labeled by dotted rectangles; the dotted red double arrow displays an improper labeling of the mitral anulus, the solid red double arrow the proper labeling) (**D**,**E**). Fourthly, identify intervening spaces between myocardial trabecula (small orange arrows) to delineate the border between compacted and non-compacted myocardium ((**F**)—4-ChV, (**G**)—2-ChV; zoom areas are labeled by dotted rectangles; the small orange arrows display the endings of the interspaces between the trabecula; the dotted white lines display an improper labeling of the compacted myocardium; the solid white line displays the proper labeling).

**Figure 2 diagnostics-13-01359-f002:**
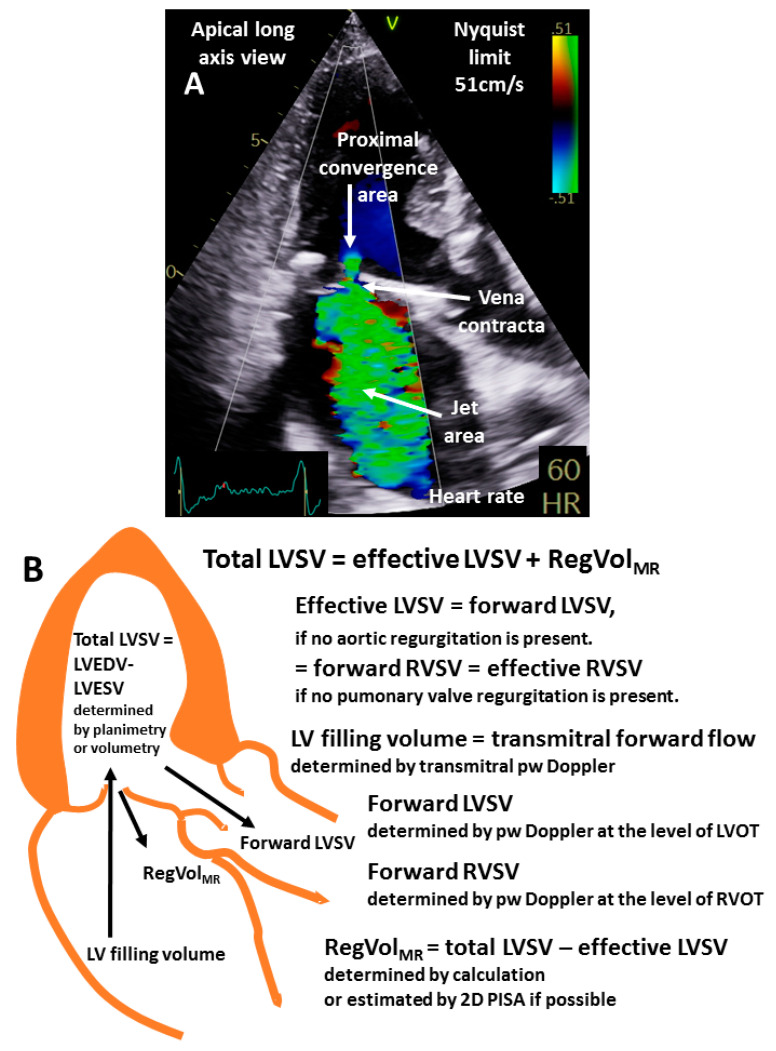
Scheme to illustrate the quantitative approach in a patient with pure mitral regurgitation (MR). The proximal convergence area, the vena contracta, and the jet area displayed in an apical long-axis view is shown in (**A**). In (**B**), a corresponding scheme is presented to explain the left ventricular (LV) and right ventricular (RV) volumes and the respective modalities for estimation by echocardiography. LVSV = LV stroke volume, RegVol_MR_ = regurgitant volume through the mitral valve, RVSV = RV stroke volume, pw = pulse wave, LVOT = left ventricular outflow tract, and RVSV = right ventricular outflow tract.

**Figure 3 diagnostics-13-01359-f003:**
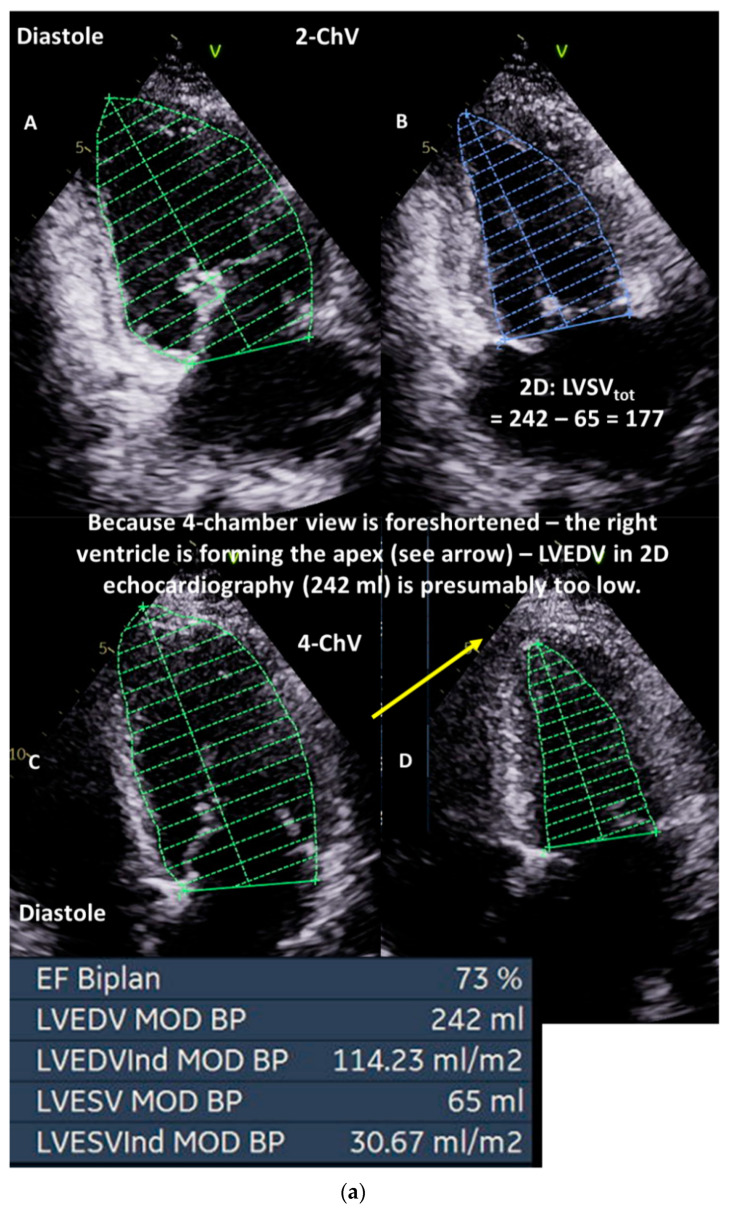
(**a**): Illustration of underestimation of LVEDV and LVESV by 2D echocardiography: in (**A**), the planimetry of LVEDV in the two-chamber view (2-ChV) is shown; in (**B**), the corresponding LVESV is shown. In (**C**), the planimetry of LVEDV in the four-chamber view (4-ChV) is shown; in (**D**), the corresponding LVESV is shown. However, the obvious difference between the longitudinal LV axis in the 4-ChV between diastole and systole indicates the foreshortening of the 4-ChV causing errors of LV volume assessment. LVSV = left ventricular stroke volume, LVSV_tot_ = total LVSV, LVSV_eff_ = effective LVSV, LVEDV = left ventricular end diastolic volume, and LVESV = left ventricular end systolic volume. (**b**): Measurements of the corresponding LVEDV areas in adjusted sectional planes by postprocessing in a 3D dataset in comparison with the 2D echocardiography presented in (**a**): in (**A**), the LVEDV assessment of the adjusted four-chamber view is shown; in (**B**), the perpendicular lines of the apical planes in the short-axis view of the 3D dataset are shown; in (**C**), the 3D view of the azimuth plane is shown; in (**D**), the LVEDV assessment of the adjusted two-chamber view is shown. In (**E**), a parasternal short-axis view during systole to label the RVOT is shown. In (**F**), the RVOT-pw-Doppler spectrum is shown. In (**G**), a parasternal long-axis view during systole to label the LVOT is shown. In (**H**), the LVOT-pw-Doppler spectrum is shown. Estimation of effective LVSV is performed with pw Doppler echocardiography by determination of forward RVSV. In isolated mitral regurgitation, a countercheck can be performed by assessment of forward LVSV which corresponds to forward RVSV. LVSV = left ventricular stroke volume, RVSV = right ventricular stroke volume, LVSV_tot_ = total LVSV, LVSV_eff_ = effective LVSV, LVEDV = left ventricular end-diastolic volume, RVOT = right ventricular outflow tract, and LVOT = left ventricular outflow tract.

## Data Availability

Data sharing not applicable.

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
