# Peer review of "Valid and Reproducible Quantitative Assessment of Cardiac Volumes by Echocardiography in Patients with Valvular Heart Diseases—Possible or Wishful Thinking?"

_diagnostics, 2023, doi:10.3390/diagnostics13071359_

Round 1

Reviewer 1 Report

Dear authors:

Thank you for the paper submitted covering an interesting topic. I think this viewpoint attemps to change the actual qualitative echocardiographic assessment of patients with valvular heart diseases (valvular regurgitations) by a definite quantitative approach.

Although the article is well written, I have some ethical concerns regarding self-citations. The first author (Andreas Hagendorff) has 7 self-citations.

I consider this amount of citations unnecessary, and its presence is not justified at all.

The figures are too small, I think they should be better presented due to this viewpoint aiming to establish a new approach in heart image acquisition. 

Figure 4 is difficult to understand. As a suggestion, they should be presented separately. 

The reference section should be updated. There are some references (seven of them) from the past century and most of them from more than 5 years.

Although the conclusions of this viewpoint are attractive, it seems that they are utopical. I think the limitations of implementations in clinical practise of this new approach should be introduced.

Author Response

Response to Reviewer1:

The authors are grateful for all comments of reviewer 1.

  1. Regarding the self-citations we understand this criticism and have reduced the citations exept the papers, in which methodological aspects of echocardiography have been described. We have eliminated the following 4 citations out of the references:
  2. Hagendorff, A.; Helfen, A.; Brandt, R.; Altiok, E.; Breithardt, O.; Haghi, D.; Knierim, J.; Lavall, D.; Merke, N.; Sinning, C.; et al. Expert Proposal to Characterize Cardiac Diseases with Normal or Preserved Left Ventricular Ejection Fraction and Symptoms of Heart Failure by Comprehensive Echocardiography. Clin Res Cardiol 2023, 112, 1–38, doi:10.1007/s00392-022-02041-y.
  3. Hagendorff, A.; Knebel, F.; Helfen, A.; Knierim, J.; Sinning, C.; Stöbe, S.; Fehske, W.; Ewen, S. Expert Consensus Document on the Assessment of the Severity of Aortic Valve Stenosis by Echocardiography to Provide Diagnostic Conclusiveness by Standardized Verifiable Documentation. Clin Res Cardiol 2020, 109, 271–288, doi:10.1007/s00392-019-01539-2.
  4. Hagendorff, A.; Doenst, T.; Falk, V. Echocardiographic Assessment of Functional Mitral Regurgitation: Opening Pandora’s Box? ESC Heart Failure 2019, 6, 678–685, doi:10.1002/ehf2.12491.
  5. Hagendorff, A.; Knebel, F.; Helfen, A.; Stöbe, S.; Doenst, T.; Falk, V. Disproportionate Mitral Regurgitation: Another Myth? A Critical Appraisal of Echocardiographic Assessment of Functional Mitral Regurgitation. Int J Cardiovasc Imaging 2021, 37, 183–196, doi:10.1007/s10554-020-01975-6.
  6. Rearrangement of all ll figures were performed and the legends were revised – especially figure 3 and 4 were devided in each 2 sub-figures to improve presentation.

See new figures in the manuscript:

Figures

Figure 1:

Instructions for proper 2D planimetry or 3D volumetry of the LV cavity. Firstly, check standardization of apical views by using triplane imaging (A). Secondly, acquire comparable monoplane 2- and 4 chamber views (2-ChV, 4-ChV) with high spatial resolution to ensure proper visualization of cardiac structures (B, C). Thirdly, delineation of mitral anulus –the proximal mitral leaflets normally do not represent the level of mitral anulus – must be properly performed (D-Zoom area is labeled by dotted rectangles; the dotted red double arrow displays an improper labeling of the mitral anulus, the solid red double arrow the proper labeling) (D, E). Fourthly, intervening spaces between myocardial trabecula (small orange arrows) must be identified to delineate the border between compacted and non-compacted myocardium (F – 4-ChV, G – 2ChV; Zoom areas are labeled by dotted rectangles; the small orange arrows display the endings of the interspaces between the trabecula; the dotted white lines display an improper labeling of the compacted myocardium, the solid white line the proper labeling).

Figure 2:

Scheme to illustrate the quantitative approach in a patient with pure mitral regurgitation (MR). The proximal convergence area, the vena contracta, and the jet area displayed in an apical long axis view is shown in A. In B a corresponding scheme is presented to explain the left ventricular (LV) and right ventricular (RV) volumes and the respective modalities for estimation by echocardiography. LVSV = LV stroke volume, RegVolMR = regurgitant volume through the mitral valve, RVSV = RV stroke volume, pw = pulse wave; LVOT = left ventricular outflow tract, RVSV = right ventricular outflow tract.

Figure 3a:

Illustration of underestimation of LVEDV and LVESV by 2D echocardiography: In A the planimetry of LVEDV in the 2-chamber view (2-ChV) is shown, in B the corresponding LVESV. In C in the planimetry of LVEDV in the 4-chamber view (4-ChV) is shown, in D the corresponding LVESV. However, the obvious difference between the longitudinal LV axis in the 4-ChV between diastole and systole indicates the foreshortening of the 4-ChV causing errors of LV volume assessment. LVSV = left ventricular stroke volume, LVSVtot = total LVSV, LVSVeff = effective LVSV, LVEDV = left ventricular end diastolic volume, LVESV = left ventricular end systolic volume

Figure 3b:

Measurements of the corresponding LVEDV areas in adjusted sectional planes by postprocessing in a 3D dataset in comparison to the 2D echocardiography presented in figure 3a: In A the LVEDV assessment of the adjusted 4-chamber view is shown, in B the perpendicular lines of the apical planes in the short axis view of the 3D dataset, in C the 3D view of the azimuth plane, in D the LVEDV assessment of the adjusted 2-chamber view  In E a parasternal short axis view during systole to label the RVOT is shown. In F the RVOT-pw-Doppler spectrum is shown. In G a parasternal long axis view during systole to label the LVOT is shown. In H the LVOT-pw-Doppler spectrum is shown.  Estimation of effective LVSV is performed with pw Doppler echocardiography by determination of forward RVSV. In isolated mitral regurgitation a countercheck can be performed by assessment of forward LVSV which correspond to forward RVSV. LVSV = left ventricular stroke volume, RVSV = right ventricular stroke volume LVSVtot = total LVSV, LVSVeff = effective LVSV, LVEDV = left ventricular end diastolic volume, RVOT – right ventricular outflow tract, LVOT – left ventricular outflow tract.

Figure 4a:

Illustration of a systolic regurgitant jet phenomenon in a patient with isolated mitral regurgitation and the quantitative analysis of left ventricular volume parameters. In A the small Vena contracta < 1mm despite a remarkable jet area is shown. In B monoplane LV planimetry is shown using the apical long axis view. In C the biplane LV planimetry is shown documenting a total LVSV of 90ml. In D a color-coded M-Mode through the MR shows a nearly constant PISA radius. In E the cw regurgitant Doppler velocity is shown. LVEDV = left ventricular end diastolic volume, LVESV = left ventricular end systolic volume, LVSV = left ventricular stroke volume, LVSVtot = total LVSV, LVEF = left ventricular ejection fraction, 

Figure 4b:

Illustration to assess quantitatively effective LVSV and RVSV by pw Doppler echocardiography as well as regurgitant volume by 2D PISA-method to analyze severity of mitral regurgitation (MR): In A the parasternal long axis to document LVOT diameter during systole is shown. In B the measurements of LVSVeff by pw Doppler spectrum at the level of the LVOT is shown, in C the parasternal and subcostal short axis views are shown to document RVOT diameter during systole. In D the measurements of RVSVeff by pw Doppler spectrum at the level of the RVOT is shown. The estimation of RegVolMR by 2D PISA is demonstrated by delineation of the 2D-PISA radius and the velocity time integral of the retrograde transmitral velocity during systole (F). LVSV = left ventricular stroke volume, LVEF = left ventricular ejection fraction, LVSVeff = effective LVSV, RegVol = transmitral regurgitant volume, LVOT – left ventricular outflow tract, RVOT – right ventricular outflow tract.

  1. Recent studies and reviews about the value of 3D echocardiography were mentioned. Thus, the reference section was updated by several citations from less than 5 years.
  2. Wu, V.; Takeuchi, M. Three-Dimensional Echocardiography: Current Status and Real-Life Applications. Acta Cardiologica Sinica 2017, 33, doi:10.6515/ACS20160818A.
  3. Lang, R.M.; Addetia, K.; Narang, A.; Mor-Avi, V. 3-Dimensional Echocardiography. JACC: Cardiovascular Imaging 2018, 11, 1854–1878, doi:10.1016/j.jcmg.2018.06.024.
  4. Baldea, S.M.; Velcea, A.E.; Rimbas, R.C.; Andronic, A.; Matei, L.; Calin, S.I.; Muraru, D.; Badano, L.P.; Vinereanu, D. 3-D Echocardiography Is Feasible and More Reproducible than 2-D Echocardiography for In-Training Echocardiographers in Follow-up of Patients with Heart Failure with Reduced Ejection Fraction. Ultrasound Med Biol 2021, 47, 499–510, doi:10.1016/j.ultrasmedbio.2020.10.022.
  5. Mancuso, F.J.N.; Moises, V.A.; Almeida, D.R.; Poyares, D.; Storti, L.J.; Brito, F.S.; Tufik, S.; de Paola, A.A.V.; Carvalho, A.C.C.; Campos, O. Prognostic Value of Real-Time Three-Dimensional Echocardiography Compared to Two-Dimensional Echocardiography in Patients with Systolic Heart Failure. Int J Cardiovasc Imaging 2018, 34, 553–560, doi:10.1007/s10554-017-1266-0.
  6. Muraru, D.; Baldea, S.M.; Genovese, D.; Tomaselli, M.; Heilbron, F.; Gavazzoni, M.; Radu, N.; Sergio, C.; Baratto, C.; Perelli, F.; et al. Association of Outcome with Left Ventricular Volumes and Ejection Fraction Measured with Two- and Three-Dimensional Echocardiography in Patients Referred for Routine, Clinically Indicated Studies. Front. Cardiovasc. Med. 2022, 9, 1065131, doi:10.3389/fcvm.2022.1065131.
  7. The consideration of limitations to implement this new approach in clinical practice were implemented into the text by the following paragraph:

The practical limitations in clinical practice are obvious to introduce a verifiable quantitative approach of cardiac volume assessment. However, this challenge must be accepted to improve echocardiography diagnostics – especially in valvular heart disease.

Reviewer 2 Report

The manuscript provides a critical and well documented analysis of the current limitations and possible improvements of ventricular volume assessment by echocardiography, especially for the evaluation of valvular regurgitation.

Minor comments

‘Questions …’ section

‘Assuming that cardiac volumes have been underestimated by echocardiography in the past, what are the clinical implications?’ – perhaps of interest for the reader would be to address the question at the present tense: Assuming that cardiac volumes are underestimated by echocardiography, what are the clinical implications?

Case report – please underline more specifically the reliability of echography volume measurements

Page 6, 3rd paragraph (#Lancellotti 2013, #Zoghbi 2017) and 5th (last) paragraph (#Lancellotti 2013) – please delete these unformatted references

Please insert numbers for the sections/ subsections of the manuscript

Author Response

Response to Reviewer2:

The authors are grateful for all comments of reviewer 2.

  1. ‘Assuming that cardiac volumes have been underestimated by echocardiography in the past, what are the clinical implications?’ – perhaps of interest for the reader would be to address the question at the present tense: Assuming that cardiac volumes are underestimated by echocardiography, what are the clinical implications?

The text was changed according to this comment.

  1. Case report – please underline more specifically the reliability of echography volume measurements.

The following phrase was added with respect to the comment:

The reliability of the echocardiographic volume measurements in this case of a pure mitral regurgitation can be counterchecked by the determination of effective LVSV and RVSV. In addition, the total LVSV must be the sum of effective LVSV and RegVolMR. The calculated RegVolMR can be checked in this case by the determination of RegVolMR by the 2D PISA-method.

  1. Page 6, 3rdparagraph (#Lancellotti 2013, #Zoghbi 2017) and 5th (last) paragraph (#Lancellotti 2013) – please delete these unformatted references

The unformatted references were deleted.

  1. Please insert numbers for the sections/ subsections of the manuscript

Numbers for sections and subsections were inserted.
